# Mitogen-Activated Protein Kinase Kinase Kinase 1 Overexpression Disrupts Development of the Ocular Surface Epithelium

**DOI:** 10.3390/cells14120894

**Published:** 2025-06-13

**Authors:** Maureen Mongan, Bo Xiao, Antonius Christianto, Yueh-Chiang Hu, Ying Xia

**Affiliations:** 1Department of Environmental and Public Health Sciences, College of Medicine, University of Cincinnati, Cincinnati, OH 45267, USA; monganmc@ucmail.uc.edu (M.M.); xiaobo@mail.uc.edu (B.X.); achristianto87@itb.ac.id (A.C.); 2Division of Developmental Biology, Cincinnati Children’s Hospital Medical Center, Cincinnati, OH 45267, USA; yueh-chiang.hu@cchmc.org; 3Department of Pediatrics, University of Cincinnati College of Medicine, Cincinnati, OH 45267, USA

**Keywords:** MAP3K1 gain-of-function, ocular surface epithelium, eyelid closure, epithelial morphogenesis

## Abstract

Mitogen-Activated Protein Kinase Kinase Kinase 1 (MAP3K1) is a key signaling molecule essential for eyelid closure during embryogenesis. In mice, *Map3k1* knockout leads to a fully penetrant eye-open at birth (EOB) phenotype due to disrupted MAPK signaling, abnormal epithelial differentiation, and morphogenesis. To further explore the roles of MAP3K1 in ocular development, we generated a Cre-inducible gain-of-function transgenic mouse, designated as *Map3k1^TG^*, and crossed it with Lens epithelial (*Le)-Cre* mice to drive MAP3K1 overexpression in developing ocular surface epithelium (OSE). *Map3k1^TG^*;*Le-Cre* embryos exhibited ocular defects including premature eyelid closure, lens degeneration, and corneal edema. While corneal epithelial differentiation remained intact, the lens epithelium degenerated with lens formation compromised. Eyelid epithelium was markedly thickened, containing cells with aberrant keratin (K)14/K10 co-expression. Genetic rescue experiments revealed that *Map3k1^TG^*;*Le-Cre* restored eyelid closure in *Map3k1* knockout mice, whereas MAP3K1 deficiency attenuated the epithelial thickening caused by transgene expression. Mechanistically, MAP3K1 overexpression enhanced c-Jun phosphorylation in vivo and activated JNK-c-Jun, WNT, TGFβ, and Notch signaling and promoted keratinocyte proliferation and migration in vitro. These findings highlight a dose-sensitive role for MAP3K1 in regulating epithelial proliferation, differentiation, and morphogenesis during eyelid development.

## 1. Introduction

The mitogen-activated protein kinase (MAPK) pathway is a central signal transduction cascade that mediates physiological and extracellular cues for cellular responses [1]. A canonical MAPK pathway comprises three tiers: a MAPK kinase kinase (MAP3K), a MAPK kinase (MAP2K), and a MAPK, with the MAP3K determining the specificity of pathway activation [2]. MAP3K1, also known as MEKK1, is a member of the MAP3K family. It contains a conserved kinase domain as well as a structurally unique regulatory region that distinguishes it from other MAP3Ks [3]. These structural features confer a broad range of signaling activities, many of which remain incompletely understood.

In vitro studies using genetic gain- and loss-of-function approaches have identified diverse and cell type-specific roles for MAP3K1 in the regulation of cell survival or apoptosis, cytoskeletal dynamics, migration, stem cell maintenance and differentiation, and cancer metastasis [4]. In vivo, *Map3k1* knockout impairs the development of the eye, auditory system, and female reproductive tract, as well as in T-cell receptor signaling, cardiac injury response, and stem cell function [5]. Studies of knockout mice have also established the essential role of MAP3K1 in embryonic eyelid closure, a morphogenetic event conserved across mammals [6,7,8,9].

Mammalian eye development involves a transient fusion of the upper and lower lids in embryogenesis [10]. In mice, eyelid formation begins at embryonic day 11.5 (E11.5), when the surface ectoderm folds at the junction between the future conjunctiva and cornea. These folds elongate to form eyelid buds, which grow toward each other until they meet and fuse by E15.5, forming a protective barrier over the developing eye [11,12]. Failure of eyelid fusion results in the EOB phenotype, characterized by open eyelids at birth [13]. The EOB mice also exhibit diverse ocular surface and adnexal structure abnormalities, indicating that the closed eyelids protect ocular surface and facilitate eye structural organization [14,15].

*Map3k1* gene knockout leads to an autosomal recessive and fully penetrant EOB phenotype [7,8,16]. In embryonic eyelids, MAP3K1 is predominantly expressed in the epithelium, where its gene ablation disrupts JNK signaling, impairs epithelial migration, and accelerates terminal differentiation [7,17]. These disruptions in turn inhibit forward migration of the eyelid tip, culminating in failed closure thereby the EOB phenotype [18].

Beyond its developmental roles, somatic *MAP3K1* mutations are associated with a range of human diseases. While its loss-of-function mutations and copy number deletions are prevalent in multiple cancers, especially luminal breast cancer [19], the gain-of-function mutations have been reported in melanoma, breast, lung, and colorectal cancers [20,21,22,23,24,25]. More recently, germline gain-of-function variants have been implicated in 46,XY Disorders of Sexual Differentiation (DSDs), congenital defects where the chromosomal male (XY) individuals display sexual abnormalities and/or female sexual characteristics [26]. Despite these associations, an in vivo model for studying MAP3K1 hyperactivation is lacking.

To address this gap, we developed an inducible *Map3k1* transgenic mouse model (*Map3k1^TG^*). Given the well-established role of MAP3K1 in embryonic eyelid closure, we tested MAP3K1 overexpression in ocular surface epithelium by crossing the *Map3k1^TG^* mice with the *Le-Cre* mice. This study characterizes the developmental effects of MAP3K1 overexpression in vivo and lays the foundation for investigating development and diseases in the context of MAP3K1 hyperactivation.

## 2. Materials and Methods

### 2.1. Mouse Strains and Genotyping

We made a construct containing Cre-inducible, CAG promoter-driven, V5- and TurboID-tagged *Map3k1* cDNA. The V5-TurboID-MAP3K1-pA sequence was synthesized (Gene Universal) and cloned into a targeting vector containing the CAG promoter, a *loxP*-stop-*loxP* cassette, and 2.6 kb and 3.4 kb homologous arms flanking the 5′ and 3′ ends, respectively (Figure 1a). DNA was purified using the EndoFree Plasmid Kit (Qiagen, Hilden, Germany). The construct was targeted downstream of the *Col1a1* locus using CRRSPR/Cas9 technology. Briefly, donor plasmid (40 ng/μL), together with Cas9 protein (IDT, Cat. #1081061) and targeting sgRNA (30 ng/μL), was microinjected into the pronuclei of fertilized C57BL/6J oocytes. Zygotes were transferred into pseudopregnant CD-1 females (~25 embryos per recipient).

The genotyping of founder mice was performed using P2/P3 and P4/P5 primer sets, Figure 1b, and correctly targeted founders, resulting in *Col1a1^V5-Map3k1TG^* allele, denoted hereafter as *Map3k1^TG^*, being bred with C57BL/6J mice. The genomic DNA of tail biopsy was used for routine PCR genotyping. P1/P2 primers identified the *Map3k1^TG^* allele, and P1/P6 primers were used to detect the wild-type allele. All primers were synthesized by IDT, and sequences are listed in Appendix A.

*Map3k1^−/−^* mice were generated previously in-house [7]. *Le-Cre* mice were a kind gift from Dr. Ashery-Padan. All mouse lines were backcrossed to the C57BL/6J background for more than 12 generations to ensure genetic uniformity. For experimental crosses, *Map3k1^TG^* were bred with *Le-Cre* mice to induce ocular surface epithelium (OSE)-specific transgene expression. Additionally, *Map3k1^TG^*;*Le-Cre* mice were crossed with *Map3k1^−/−^* to investigate genetic rescue of the EOB phenotype.

### 2.2. Phenotype Evaluation, Histology, and Immunohistochemistry

Embryos and fetuses were harvested at specified embryonic stages, and heads were fixed overnight in 4% paraformaldehyde (PFA; Thermo Fisher Scientific, Waltham, MA, USA) at 4 °C. Gross eye morphology was documented using a Leica MZ-16FA dissecting microscope. Eyelid opening size was quantified by tracing the area enclosed by the leading edges using ImageJ v1.54g. https://imagej.net/ij/ (accessed in January 2012). Pixel values were converted to absolute measurements (mm^2^) using calibrated scale bars. Complete eyelid closure was assigned to a value of zero. All chemical reagents used in the experiments are listed in Appendix A.

Fixed tissues were processed through paraffin embedding, and serial sections (5 μm) were cut using a rotary microtome (MT-980, Research & Manufacturing Co., Inc., Tucson, AZ, USA). After deparaffinization, sections were stained with hematoxylin and eosin (H&E) and examined via bright-field microscopy (Zeiss AXIO Scope A1, Cambridge, UK). Images were acquired using Zen 3.1 (Blue edition) software.

For immunohistochemistry, deparaffinized sections underwent antigen retrieval in boiling citrate buffer (10 mM, pH 6.0) for 30 min. Sections were blocked with 5% bovine serum albumin (BSA) in PBS at room temperature for 1 h and incubated with primary antibodies overnight at 4 °C. Following secondary antibody incubation and nuclear counterstaining with Hoechst 33342, slides were imaged using a Zeiss Axioplan 2 fluorescence and confocal microscope. The primary and secondary antibodies used are detailed in Appendix A.

### 2.3. Keratinocyte Culture

Primary keratinocytes were isolated from *Map3k1^TG^* neonatal mouse skin and cultured in calcium-free Keratinocyte Serum-Free Medium (KSFM-Ca; Gibco, Waltham, MA, USA, Cat. #10725-018), as described previously [27]. Cells were infected for 48 h with adenoviruses encoding either green fluorescent protein (Ad-GFP) or GFP plus Cre recombinase (Ad-Cre) to induce transgene expression.

For pathway reporter assays, *Map3k1^TG^* keratinocytes were infected with lentiviruses encoding luciferase reporters specific to various signaling pathways (Appendix A) in the presence of 8 µg/mL polybrene. Infected cells were selected with 1 µg/mL blasticidin (Thermo Fisher, Cat. #R21001, Waltham, MA, USA) until uninfected cells were eliminated. Stable reporter cells were subsequently infected with Ad-GFP or Ad-Cre, and lysates were collected for luciferase activity assays using the Luciferase Assay System (Promega, Madison, WI, USA). Luciferase signals were normalized to total protein concentration. All experiments included at least three biological replicates.

### 2.4. Proliferation and Apoptosis

To assess in vivo proliferation, pregnant mice were injected intraperitoneally with 5-ethynyl-2′deoxyuridine (EdU, 5 mg/kg body weight) and sacrificed 2 h later. Embryos were harvested, fixed, paraffin-embedded, and sectioned as described above. For in vitro proliferation analysis, cultured *Map3k1^TG^* keratinocytes were labeled with 10 µM EdU for 2 h, fixed in 4% PFA for 30 min at 4 °C, and permeabilized with 0.2% Triton X-100.

EdU incorporation was detected using the iClick EdU Imaging Kit (ABP Biosciences, Rockville, MD, USA), according to the manufacturer’s protocol. Apoptosis was assessed using the ApopTag Peroxidase In Situ Apoptosis Detection Kit (Millipore Sigma, Burlington, MA, USA). Images were acquired with a Zeiss Axio microscope. For each condition, positively stained cells were quantified from three sections per embryo, with at least three embryos analyzed per genotype.

### 2.5. Wound Healing Assay

*Map3k1^TG^* keratinocytes infected with Ad-GFP and Ad-Cre for 24 h were cultured to confluence and then subjected to a linear scratch using a sterile pipette tip. Images were captured immediately (0 h) and at 24 h post-scratch using a Zeiss Axioplan 2 fluorescence microscope. The wound area was measured using the Wound Healing Size Tool in ImageJ, and the percentage of wound closure was calculated using Equation (1) [28]:(1)$$Wound\left(\%\right)=\frac{A24}{A0}\times 100\%$$
where A0 is the initial area of the scratch circle, and A24 is the transparent area after culturing for 24 h.

### 2.6. Western Blot Analyses

Keratinocytes were lysed in RIPA buffer containing 20 mM Tris-HCl (pH 7.5), 150 mM NaCl, 1 mM EDTA, 1% NP-40, 0.5% sodium deoxycholate, and protease/phosphatase inhibitors (10 µg/mL aprotinin, 10 µg/mL leupeptin, 1 mM Na_3_VO_4_, and 1 mM PMSF). Protein samples (30–50 µg) were separated by SDS-PAGE and transferred to nitrocellulose membranes. Membranes were probed with primary and HRP-conjugated secondary antibodies (Appendix A), and signals were detected using a UVP Biochemi System (G58-bc-042704). Band intensities were quantified with ImageJ densitometry, normalized to β-actin as a loading control. The background signal was subtracted, and relative expression was calculated by dividing the adjusted protein signal by the β-actin signal and normalizing it to the control group (set as 1).

### 2.7. Statistical Analysis

All experiments were conducted with at least three independent biological replicates unless otherwise noted. Data are presented as mean ± s.e.m. Statistical comparisons were performed using Student’s two-tailed *t*-tests or one-way analysis of variance following Dunnett’s test for multiple comparisons. Sample sizes and specific statistical methods are indicated in the corresponding figure legends. Statistical significance was defined as follows: *p* < 0.05 (*), *p* < 0.01 (**), and *p* < 0.001 (***).

## 3. Results

### 3.1. MAP3K1 Overexpression in OSE: Transgenic Mouse Model Characterization

To model MAP3K1 gain-of-function in vivo, a transgenic mouse line *Map3k1^TG^* was made, carrying a Cre-inducible *Map3k1* transgene inserted into the safe harbor *Col1a1* locus using CRISPR/Cas9 [29] (Figure 1a). The transgenic construct consisted of a CAG promoter, loxP-flanked STOP cassette, and a V5- and TurboID-tagged *Map3k1* cDNA. PCR genotyping confirmed correct integration of the transgene Figure 1b.

To drive ocular surface epithelium (OSE)-specific expression, *Map3k1^TG^* mice were crossed with *Le-Cre* mice. The immunostaining of E15.5 eyes revealed the robust expression of V5-MAP3K1 in the epithelial layers of the eyelid, cornea, and lens in *Map3k1^TG/+^*;*Le-Cre* embryos, but not in *Map3k1^TG/+^* controls lacking Cre (Figure 1c). V5-MAP3K1 was most prominent in suprabasal epithelial cells and lens fibers, with relatively lower levels in basal epithelial cells. Retinal and stromal compartments lacked detectable expression, confirming OSE specificity of transgene expression.

High-magnification imaging of suprabasal eyelid cells further showed that V5-MAP3K1 localized in punctate clusters within the cytoplasm and on the plasma membrane (Figure 1d), suggesting potential involvement in signaling microdomains or super complexes.

### 3.2. MAP3K1 Overexpression Alters Corneal, Lens, and Eyelid Development

To assess the developmental effects of MAP3K1 overexpression, *Map3k1^TG/+^* and *Map3k1^TG/+^*;*Le-Cre* embryonic eyes at E15.5 and E17.5 were analyzed histologically. In both genotypes, the corneal epithelium was composed of two well-aligned cell layers above the stroma (Figure 2a,b). Immunostaining at E17.5 revealed that basal cells expressed keratin 14 (K14), and suprabasal cells expressed keratin 12 (K12), a corneal epithelial marker, indicating proper epithelial stratification and differentiation (Figure 2c). These findings suggest that MAP3K1 overexpression does not impair corneal epithelial specification.

Despite normal epithelial differentiation, the *Map3k1^TG/+^*;*Le-Cre* corneas exhibited posterior segment abnormalities, including increased fibrosis in corneal stroma (α-smooth muscle actin, α-SMA) and tight junction proteins (Zonula Occludens-1, ZO-1) in Descemet’s membrane (Figure 2d), likely secondary to anterior lens degeneration. In contrast to the well-organized, cuboidal lens epithelium observed in *Map3k1^TG^* embryos, *Map3k1^TG/+^*;*Le-Cre* embryos showed severe epithelial thinning and structural disorganization by E15.5, progressing to complete loss of lens epithelium by E17.5 (Figure 2a,e). Lens fibers also appeared degenerated.

Quantification confirmed a significant reduction (*p* < 0.001) in the number of lens epithelial cells in *Map3k1^TG/+^*;*Le-Cre* embryos compared to the controls (Figure 2f). Proliferation assays using EdU labeling showed comparable mitotic rates between genotypes in both corneal and lens epithelia (Figure 2g). Similarly, apoptosis levels, measured by TUNEL staining, were slightly elevated in the *Map3k1^TG/+^*;*Le-Cre* lens but did not reach statistical significance (Figure 2h). These data suggest that lens epithelial loss may occur prior to E15.5, potentially also impairing corneal endothelial development and contributing to corneal edema (Figure 2c).

The embryonic eyelids showed complete fusion by E17.5 in both genotypes, consistent with the normal developmental timeline (Figure 2a). However, at E15.5, *Map3k1^TG/+^*;*Le-Cre* embryos had significantly smaller (*p* < 0.01) open-eye area than the controls (Figure 3a,b), suggesting accelerated eyelid closure. The eyelid epithelium of E15.5 transgenic embryos exhibited additional abnormalities. The control epithelium was organized into two distinct layers: K14-positive basal and K10-positive suprabasal cells (Figure 3c–e). In contrast, *Map3k1^TG/+^*;*Le-Cre* eyelids displayed a markedly thickened epithelium containing additional cell layers and a population of K14/K10 double-positive cells, a hallmark of transient amplifying keratinocytes with high self-renewal capacity [30,31,32]. Correspondingly, EdU incorporation assays revealed significantly increased (*p* < 0.001) proliferation in the *Map3k1^TG/+^*;*Le-Cre* eyelid epithelium, while apoptosis remained only modestly elevated and not statistically significant (Figure 3f). These findings indicate that MAP3K1 overexpression disrupts normal stratification and homeostasis of the eyelid epithelium, promoting expansion of a proliferative keratinocyte population.

### 3.3. MAP3K1 Expression Level Correlates with Severity of Ocular Defects

To determine whether the severity of ocular abnormalities is influenced by MAP3K1 expression level, *Map3k1^TG/TG^*;*Le-Cre* embryos, which carry two transgene alleles, were examined. Compared to heterozygotes, homozygous embryos exhibited markedly stronger V5-MAP3K1 expression in OSE tissues (Figure 4a). As in *Map3k1^TG/+^*;*Le-Cre* embryos, the overexpressed protein localized as clusters and had a broader cytoplasmic distribution in suprabasal eyelid epithelial cells (Figure 4b).

The *Map3k1^TG/TG^*;*Le-Cre* embryos displayed more severe ocular abnormalities. The lens epithelium was completely degenerated by E15.5, and the lens was entirely absent at E17.5 (Figure 4c), indicating a more profound disruption of lens development than in heterozygotes. The corneal epithelium in these embryos lacked both K14 and K12 expression (Figure 4d), suggesting a loss of epithelial identity and/or failure of differentiation.

As with *Map3k1^TG/+^*;*Le-Cre* embryos, eyelid closure was accelerated in homozygotes, and the eyelid epithelium was considerably thickened (Figure 4e,f). However, in contrast to heterozygotes, *Map3k1^TG/TG^*;*Le-Cre* embryos exhibited a substantial increase in apoptosis within the eyelid epithelium, as shown by TUNEL staining (Figure 4g,h). These findings indicate that excessive MAP3K1 expression surpasses a threshold that triggers pro-apoptotic signaling in OSE cells, further impairing tissue homeostasis.

### 3.4. MAP3K1 Overexpression Rescues the Eye-Open at Birth Phenotype in Knockout Mice

To determine whether MAP3K1 overexpression could functionally compensate for MAP3K1 deficiency, genetic rescue experiments were conducted by crossing *Map3k1^TG/+^*;*Le-Cre* mice with *Map3k1^−/−^* mutants. As expected, *Map3k1^−/−^* fetuses displayed the characteristic eye-open at birth (EOB) phenotype due to failed eyelid closure (Figure 5a). The *Map3k1^−/−^*;*Map3k1^TG/+^*;*Le-Cre* fetuses, on the other hand, exhibited closed eyelids, indicating effective rescue of the EOB phenotype (Figure 5a,b). The compound mutants had thickened eyelid epithelium; however, their epithelial thickness was significantly reduced (*p* < 0.001) compared to *Map3k1^TG/+^*;*Le-Cre* embryos alone (Figure 5c,d). Thus, the absence of endogenous MAP3K1 partially alleviated the hyperproliferative effects of transgenic MAP3K1 expression. These findings demonstrate that MAP3K1 gain-of-function can rescue the eyelid closure defect caused by MAP3K1 deficiency and that precise MAP3K1 dosage is important in maintaining epithelial homeostasis during ocular development.

### 3.5. MAP3K1 Overexpression Activates JNK, WNT, TGFβ, and Notch Signaling Pathways

To explore the signaling pathways activated by MAP3K1 overexpression, keratinocyte lines from *Map3k1^TG^* mice were established. Adenoviral Cre (Ad-Cre) infection induced strong V5-MAP3K1 expression in these cells, while GFP-only infection (Ad-GFP) did not (Figure 6a). Western blot analysis showed that MAP3K1 induction led to the robust activation of downstream MAPK components, including phospho-JNK, phospho-p38, phospho-ERK, and the JNK substrate phospho-c-Jun (Figure 6a,b). Functionally, Ad-Cre-induced *Map3k1^TG^* keratinocytes displayed significantly increased proliferation (*p* < 0.01) (EdU labeling) and accelerated wound closure (*p* < 0.01) in scratch assays compared to Ad-GFP controls (Figure 6c–e).

To further investigate the effects of MAP3K1 overexpression on signaling pathways, stable *Map3k1^TG^* keratinocytes harboring luciferase reporters for specific signaling/transcriptional cascades were generated. Cre-mediated MAP3K1 overexpression considerably increased AP-1 (JNK/c-Jun), TCF (WNT), BRE (TGFβ), and CBF1/RBPJ (Notch) reporter activity, relative to the GFP controls (Figure 6f). Importantly, no changes were observed in serum response element (SRE) reporter activity, indicating selective pathway activation. In vivo*,* the immunostaining of E15.5 eyelids showed elevated phospho-c-Jun expression specifically at the epithelial leading edge in *Map3k1^TG/+^*;*Le-Cre* embryos, but not in *Map3k1^TG/+^*controls (Figure 6g). This finding supports the role of MAP3K1 in activating JNK/c-Jun signaling at key morphogenetic sites. Collectively, our data demonstrate that MAP3K1 overexpression activates multiple developmental signaling pathways, promoting epithelial proliferation and migration, and disrupting the balance of differentiation and apoptosis in a dose-dependent and context-specific manner.

## 4. Discussion

Although MAP3K1 is well known for its crucial role in embryonic eyelid closure, the molecular and cellular mechanisms it governs during this process remain poorly understood. In this study, we used a forward genetic approach to investigate the consequences of MAP3K1 overexpression in vivo. Utilizing a novel inducible transgenic model (*Map3k1^TG^*), we demonstrate that MAP3K1 gain-of-function in the ocular surface epithelium (OSE) leads to accelerated eyelid closure, disrupted epithelial homeostasis, and aberrant differentiation and dose-dependent pathogenesis of the eyelid, cornea, and lens.

Our data show that transgene induction drives the abnormal expansion of the eyelid epithelium, which includes not only the typical basal (K14-positive) and suprabasal (K10-positive) keratinocyte layers but also a unique population of K14/K10 double-positive cells. In the epidermis, K14 marks proliferative basal keratinocytes, while K10 is expressed in differentiating suprabasal cells during upward stratification [33]. The presence of K14/K10 double-positive cells, often referred to as transient amplifying keratinocytes, reflects a population poised for limited proliferation and early differentiation [34,35]. In *Map3k1^TG^*;*Le-Cre* embryos, sustained K14 expression in cells already expressing K10 suggests a disruption in the normal transition to terminal differentiation. This alteration was accompanied by elevated cell proliferation, indicating that MAP3K1 overexpression may lock cells in a proliferative and partially differentiated state. The phenotypic changes align with prior findings in loss-of-function studies, reinforcing the view that MAP3K1 is a critical regulator of epithelial stratification and morphogenesis [17]. While the knockout models display reduced JNK activity and impaired eyelid tip migration, our overexpression model reveals the opposite effect, i.e., enhanced proliferation, accelerated migration, and premature closure, highlighting a dosage-sensitive role for MAP3K1 in epithelial dynamics.

At the molecular level, our data show that MAP3K1 overexpression activates multiple signaling pathways, including JNK/c-Jun, WNT, TGFβ, and Notch. These pathways have distinct and well-established roles in regulating keratinocyte behavior. For example, c-Jun phosphorylation through JNK promotes AP-1 activity, which is known to drive *K14* gene expression and influence keratinocyte proliferation [36]. TGFβ and WNT signaling can regulate stem/progenitor cell expansion and differentiation programs [37,38,39]. Notch signaling is particularly important for the transition from basal to suprabasal keratinocytes during early epidermal stratification [40,41]. Together, these signaling cascades likely coordinate to produce the expanded, disorganized, and hyperproliferative epithelial phenotype observed in *Map3k1^TG^*;*Le-Cre* embryos.

While the corneal epithelium appeared to retain its differentiation capacity (e.g., expression of K14 and K12), MAP3K1 overexpression severely disrupted lens development. Lens epithelial cells were lost early during development, and lens fibers failed to properly form. These effects occurred without detectable changes in proliferation or apoptosis at E15.5, suggesting that MAP3K1 disrupts key lens developmental processes prior to this stage. Lens degeneration also coincided with defects in the overlaying corneal endothelium and Descemet’s membrane, likely through non-cell-autonomous mechanisms such as altered signaling or loss of trophic support. The varying outcomes across different cell types in our model suggest that MAP3K1 activates both shared and tissue-specific programs. For instance, c-Jun is essential for eyelid closure but dispensable for corneal epithelial development [42]. This context specificity reflects the differential expression of cofactors or variations in signal integration among distinct OSE-derived cell types.

From a functional standpoint, the gain-of-function model also provided mechanistic insight into MAP3K1 dosage sensitivity. While transgene expression from a single allele was sufficient to rescue the eye-open at birth (EOB) phenotype in *Map3k1^−/−^* embryos, overexpression from two alleles resulted in severe epithelial disorganization, loss of differentiation, and apoptosis. These findings highlight the importance of tightly regulated MAP3K1 expression during development. Subcellular localization also varied with expression level: at moderate levels, MAP3K1 formed puncta in suprabasal epithelial cells, while higher expression led to partial redistribution to the cytosol and membrane, possibly altering interaction networks or triggering pro-apoptotic pathways [43,44,45].

Collectively, this study shows that MAP3K1 functions as a dose-sensitive regulator of epithelial proliferation, migration, and differentiation. The *Map3k1^TG^* model represents a powerful tool for investigating developmental and disease-related processes driven by MAP3K1 hyperactivation, including cancer, congenital disorders, and epithelial pathologies.

## Figures and Tables

**Figure 1 cells-14-00894-f001:**
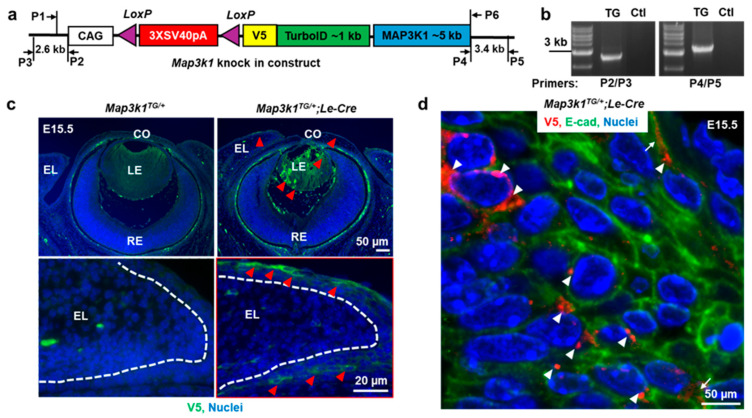
Characterization of the *Map3k1^TG^* mouse model for OSE-specific overexpression. (**a**) Schematic diagram of the transgenic construct showing the CAG promoter, *loxP*-flanked STOP cassette, and V5- and TurboID-tagged *Map3k1* cDNA, with primer positions indicated. (**b**) Genotyping PCR of genomic DNA from control (Ctl) and transgenic (TG) mice. (**c**) Immunofluorescence of E15.5 eye sections showing V5-MAP3K1 expression (green) in *Map3k1^TG/+^*;*Le-Cre* but not *Map3k1^TG/+^* embryos. Nuclei were counterstained with Hoechst (blue). Dash lines mark basement membrane of the eyelid epithelium; red arrowheads point at V5-MAP3K1-positive staining. (**d**) High-magnification image of V5-MAP3K1 (red) in suprabasal eyelid epithelial cells, co-stained with E-cadherin (green). Punctate V5 signal localizes to cytoplasm (arrowheads) and plasma membrane (arrows). Abbreviations: CO: cornea; LE: lens; RE: retina; EL: eyelid.

**Figure 2 cells-14-00894-f002:**
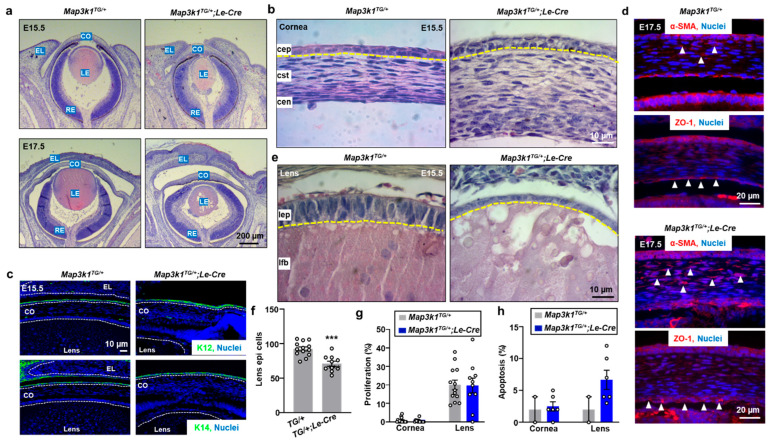
MAP3K1 overexpression alters corneal and lens development. (**a**) H&E staining of E15.5 and E17.5 eye sections from *Map3k1^TG/+^* and *Map3k1^TG/+^*;*Le-Cre* embryos/fetuses and images were captured at low magnification. (**b**,**e**) High magnification views of cornea and lens. (**c**,**d**) Immunostaining for K12 (green, suprabasal) and K14 (green, basal) in epithelium, ZO-1 in endothelium, and α-SMA in stroma of E15.5 corneas. Arrowheads point at staining positive cells. (**f**) Quantification of lens epithelial cell number. (**g**) EdU incorporation for cell proliferation. (**h**) TUNEL-positive apoptotic cells. Data represent mean ± s.e.m. from ≥3 sections per embryo, with ≥3 embryos per genotype. *** *p* < 0.001. Abbreviations: CO, cornea, EL, eyelid, RE, retina, LE, lens, cep, corneal epithelium; cst, stroma; cen, endothelium; lep, lens epithelium; lfb, lens fibers. Dash lines mark the basement membrane of epithelium.

**Figure 3 cells-14-00894-f003:**
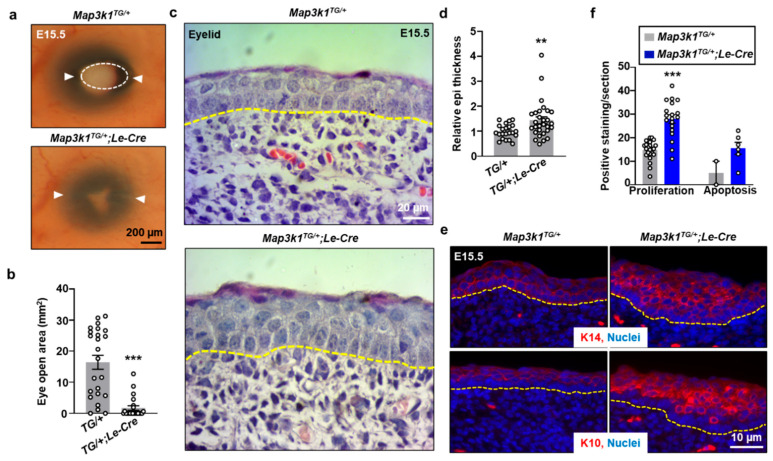
MAP3K1 overexpression affects eyelid morphogenesis. (**a**) Gross images of E15.5 embryonic eyes showing smaller eye openings in *Map3k1^TG/+^*;*Le-Cre*. White arrowheads mark canthi of the eyelids. (**b**) Quantification of open-eye area. (**c**) H&E staining of eyelid epithelium. (**d**) Measurement of epithelial thickness. (**e**) Immunostaining for K14 (top) and K10 (bottom). (**f**) Quantification of EdU-positive and TUNEL-positive cells. Dash lines mark basement membrane of eyelid epithelium. Data represent mean ± s.e.m. from ≥3 sections and ≥3 embryos per genotype. ** *p* < 0.01; *** *p* < 0.001.

**Figure 4 cells-14-00894-f004:**
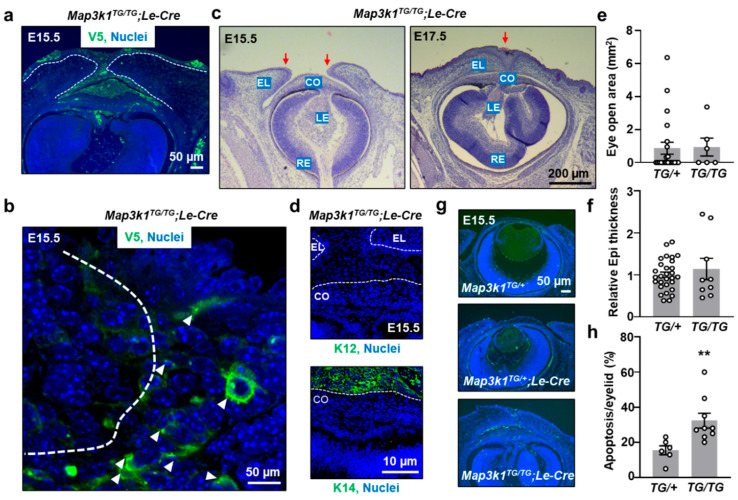
Severity of eye defects correlates with MAP3K1 expression level. (**a**,**b**) Immunostaining for V5-MAP3K1 in *Map3k1^TG/TG^*;*Le-Cre* E15.5 eyelids. Arrowheads point at V5-positive signals. (**c**) H&E staining of eye sections at E15.5 and E17.5. Red arrows mark eyelid leading edge. (**d**) K12 and K14 immunostaining of corneal epithelium. (**e**) Quantification of open-eye area. (**f**) Eyelid epithelium thickness. (**g**) TUNEL staining and (**h**) quantification of apoptotic cells. Dash lines mark basement membrane of the epithelium. Data represent mean ± s.e.m. from ≥3 sections and ≥3 embryos per genotype. ** *p* < 0.01.

**Figure 5 cells-14-00894-f005:**
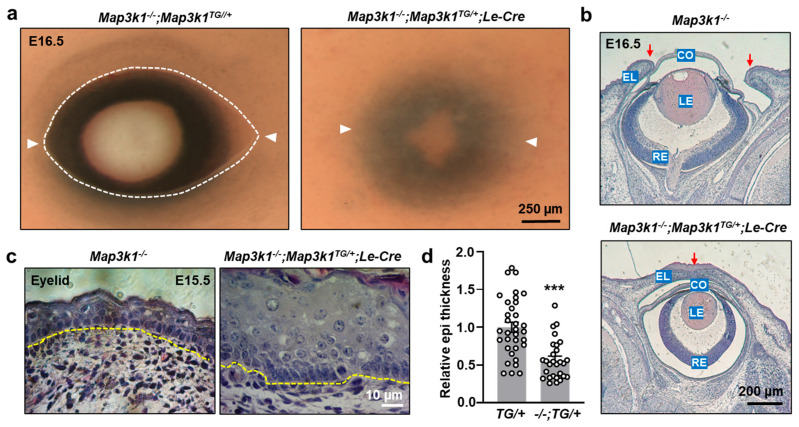
MAP3K1 overexpression rescues the EOB phenotype in *Map3k1^−/−^* mice. (**a**) Gross eye images of *Map3k1^−/−^*;*Map3k1^TG/+^* and *Map3k1^−/−^*;*Map3k1^TG/+^*;*Le-Cre* fetuses at E16.5. White arrowheads mark canthi of the eyelids. (**b**) H&E staining and (**c**) high-magnification images of eyelid epithelium. Yellow dash lines mark basement membrane of eyelid epithelium. Red arrows point at eyelid leading edge/fusion junction. (**d**) Quantification of epithelial thickness. Data represent mean ± s.e.m. from ≥3 sections and ≥3 embryos per genotype. *** *p* < 0.001. Abbreviations: CO: cornea; LE: lens; RE: retina; EL: eyelid.

**Figure 6 cells-14-00894-f006:**
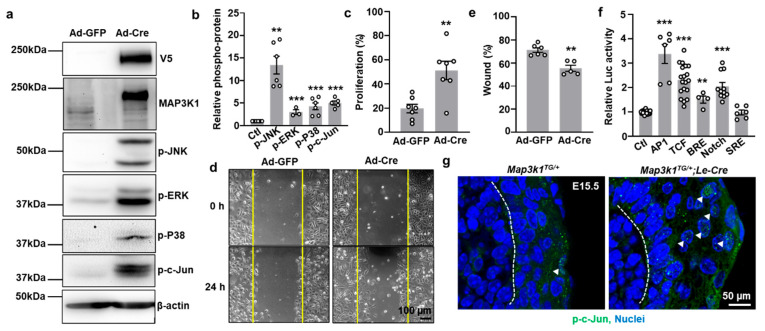
MAP3K1 overexpression activates multiple signaling pathways in keratinocytes. (**a**) Western blots detecting V5-MAP3K1, endogenous MAP3K1, p-JNK, p-ERK, p-p38, p-c-Jun, and β-actin in Ad-GFP- or Ad-Cre-infected cells. (**b**) Densitometric quantification of phospho-proteins. (**c**) Proliferation measured by EdU incorporation. (**d**) Representative images of wound healing assays in Ad-GFP- and Ad-Cre-infected cells and (**e**) quantification of wound areas at 0 and 24 h of wounding. (**f**) AP-1, TCF, BRE, CBF1/RBP, and SRE reporter activities. (**g**) Immunostaining for p-c-Jun (green, arrowheads) in E15.5 eyelid epithelium; Hoechst (blue) labels nuclei and dash lines mark basement membrane. Data represent mean ± s.e.m. from ≥3 biological replicates. ** *p* < 0.01; *** *p* < 0.001.

## Data Availability

The data presented in this study are available from the corresponding author on reasonable request.

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
