# Peer review of "Mitogen-Activated Protein Kinase Kinase Kinase 1 Overexpression Disrupts Development of the Ocular Surface Epithelium"

_cells, 2025, doi:10.3390/cells14120894_

Round 1
Reviewer 1 Report
Comments and Suggestions for Authors
The current manuscript aims to report that mitogen-activated protein kinase kinase kinase 1 overexpression disrupts development of the ocular surface epithelium. Although the topic is interesting in its scientific field, there are some issues that require the authors’ attention to improve the quality of this particular manuscript before further consideration for publication in a high-quality journal “Cells”.
Specific comments:
- Please perform immunostaining of corneal endothelial markers such as ZO-1 to clarify whether MAP3K1 overexpression can impair the endothelial function.
- The focus of this study is the analysis of specific markers and phosphorylated protein expressions. Please also provide RNA-seq data to understand the transcriptional changes caused by MAP3K1 overexpression.
- Although this manuscript shows the results of quantitative analysis, the authors should present the qualitative imaging data of the wound at the initial stage and 24 h after healing.
- Please identify the scientific meaning of white arrows shown in Figures 3a and 4b.
- Please perform the staining of fibrosis-related markers to verify the occurrence of fibrosis (rather than simple tissue remodeling).
- As mentioned in Section 2.5 ‘Wound healing assay’, the wound area was measured to calculate the percentage of wound closure. Nevertheless, in my opinion, this important experimental claim about the the equation is not supported by any appropriate documentation. If possible, please consider the inclusion of the following relevant case study (DOI: 10.1016/j.mtbio.2025.101713) in the reference list to strengthen manuscript quality and support the use of wound area-based formula.
Author Response
Comment-1: Please perform immunostaining of corneal endothelial markers such as ZO-1 to clarify whether MAP3K1 overexpression can impair the endothelial function.
Answer-1: We conducted additional immunostaining and detected a slight increase of α-SMA in corneal stroma and ZO-1 in corneal endothelium of Map3k1TG embryos. These results are included as a new Fig. 2d. As corneal stroma and endothelium are not direct targets of MAP3K1 overexpression in Map3k1TG;Le-Cre mice, the above abnormalities are considered secondary to primary defects of corneal and/or lens epithelium due to MAP3K1 overexpression.
Comment-2: The focus of this study is the analysis of specific markers and phosphorylated protein expressions. Please also provide RNA-seq data to understand the transcriptional changes caused by MAP3K1 overexpression.
Answer-2: The primary objective of this manuscript is to establish and characterize a novel Map3k1 transgenic mouse model. We aim to provide a foundational tool for the research community to explore the temporal, spatial, and cell-type–specific effects of MAP3K1 overactivation in various tissues. While transcriptomic analysis of MAP3K1-induced changes is a compelling direction for future research, it is beyond the current scope of this initial characterization study.
Comment-3: Although this manuscript shows the results of quantitative analysis, the authors should present the qualitative imaging data of the wound at the initial stage and 24 h after healing.
Answer-3: Representative images of wound healing assay at the initial stage and 24 hours post-injury have now been included as a new Fig. 6d.
Comment-4: Please identify the scientific meaning of white arrows shown in Figures 3a and 4b.
Answer-4: Figure legends have been revised to indicate the meaning of the arrows.
Comment-5: Please perform the staining of fibrosis-related markers to verify the occurrence of fibrosis (rather than simple tissue remodeling).
Answer-5: As in responses to Comment-1, new data, showing minor increase of fibrotic marker in Map3k1TG;Le-Cre corneal stroma, are included as Fig. 2d.
Comment-6: As mentioned in Section 2.5 ‘Wound healing assay’, the wound area was measured to calculate the percentage of wound closure. Nevertheless, in my opinion, this important experimental claim about the the equation is not supported by any appropriate documentation. If possible, please consider the inclusion of the following relevant case study (DOI: 10.1016/j.mtbio.2025.101713) in the reference list to strengthen manuscript quality and support the use of wound area-based formula.
Answer-6: The equation used to calculate wound closure has been revised in the Materials and Methods section based on the reference recommended by the reviewer. This citation has been added to the reference list. The updated analysis (Fig. 6e) remains consistent with our conclusion that MAP3K1 induction accelerates wound closure.
Reviewer 2 Report
Comments and Suggestions for Authors
The manuscript “MAP3K1 Overexpression Disrupts Development of the Ocular Surface Epithelium” is focused on the roles of MAP3K1 in ocular development. Authors generated transgenic mouse, to drive MAP3K1 overexpression in developing ocular surface epithelium. They demonstrated a dose-sensitive role of MAP3K1 in regulating epithelial proliferation, differentiation, and morphogenesis during eyelid development. The work was performed at a high experimental level. The manuscript makes a good impression and can be published after minor remarks are corrected. Thus, the article meets the criteria for “Minor revision”.
The Abstract is written with a clear problem statement and the results are adequately described.
The Introduction section describes the role of MAP3K1 in ocular embryogenesis, outlining abnormalities associated with MAP3K1 loss-of-function and with gain of MAP3K1. Please justify briefly why exactly Map3k1TG mice with Le-Cre mice crossbreeding was used?
Materials and Methods section
Section “2.1. Mouse strains and genotyping” is written too shorter. I recommend that Part 2.1 be divided into subchapters with additional information that will allow reproduction of the methodology. In particular, the generation of genetic constructs can be described independently; the age of mice, their maintenance, and the procedure for obtaining pseudopregnant CD-1 females can be also described. Please, indicate what equipment was used for microinjection.
Please, indicate the source of genetic material isolation for routine genotyping analysis, as well as what type of analysis was performed using primers P1/P2 and P1/P6.
Results section
1) What proteins or regulatory elements encode V5- and TurboID cDNAs? Please describe in this part (3.1.) or in the Methods section.
2) Fig. 1a-b: Can a full-length insertion assay (PCR with primers P1/P6) be added?
Discussion section
The Discussion section covers the main points of the paper and provides all the necessary references.
Author Response
Comment-1: Please justify briefly why exactly Map3k1TG mice with Le-Cre mice crossbreeding was used?
Answer-1: The Introduction has been revised, specifically on page 4, lines 75-76, to clarify the rationale for using Map3k1TG;Le-Cre mice.
Comment-2: Section “2.1. Mouse strains and genotyping” is written too shorter. I recommend that Part 2.1 be divided into subchapters with additional information that will allow reproduction of the methodology. In particular, the generation of genetic constructs can be described independently; the age of mice, their maintenance, and the procedure for obtaining pseudopregnant CD-1 females can be also described. Please, indicate what equipment was used for microinjection.
Answer-2: This section has been revised to improve clarity.
Comment-3: Please, indicate the source of genetic material isolation for routine genotyping analysis, as well as what type of analysis was performed using primers P1/P2 and P1/P6.
Answer-3: Genomic DNA isolated from tail biopsies was used for routine PCR-based genotyping with primers P1/P2 and P1/P6. This information has now been clarified in the Materials and Methods section.
Comment-4: What proteins or regulatory elements encode V5- and TurboID cDNAs? Please describe in this part (3.1.) or in the Methods section.
Answer-4: The fusion protein V5-TurboID-MAP3K1 is regulated by a CAG promoter.
Comment-5: Fig. 1a-b: Can a full-length insertion assay (PCR with primers P1/P6) be added?
Answer-5: The full-length insertion exceeds 8 kb, which makes it unsuitable for routine PCR validation due to technical limitations of standard PCR amplification.
Reviewer 3 Report
Comments and Suggestions for Authors
1- Abbreviations should be avoided in the abstract for better clarity.
2- The Materials section appears to be missing and should be included.
3- The design of the review study is not clearly described or emphasized.
4- Ethical committee approval is not mentioned and should be reported if applicable.
5- The manuscript should be written in the passive voice to maintain a formal academic tone.
6- The conclusions and recommendations are missing and need to be clearly stated.
Author Response
Comment-1: Abbreviations should be avoided in the abstract for better clarity.
Answer-1: In consideration of the length limitations, we have retained only well-recognized abbreviations (e.g., c-Jun, JNK, WNT) without definitions, while less common abbreviations, such as Le-Cre and Keratin, have been spelled out to improve clarity.
Comment-2 and 3: The Materials section appears to be missing and should be included; The design of the review study is not clearly described or emphasized.
Answers-2 & 3: It is possible that the reviewer did not receive the complete version of the submission. We confirm that the Materials section and study design details were included in the original manuscript. We would be happy to provide any additional information or clarification as needed upon the reviewer’s request.
Comment-4: Ethical committee approval is not mentioned and should be reported if applicable.
Answer-4: The ethical approval information is provided in the “Institutional Review Board Statement” section of the manuscript and/or has been submitted to the editorial office as required.
Comment-5: The manuscript should be written in the passive voice to maintain a formal academic tone.
Answers-5: Revisions have been made to improve the academic tone by incorporating passive voice where appropriate.
Comment-6: The conclusions and recommendations are missing and need to be clearly stated.
Answers-6: The conclusions and recommendations can be found in the final sentence of Abstract and the last paragraph of Discussion section.
Round 2
Reviewer 1 Report
Comments and Suggestions for Authors
The revised version has adequately addressed most of the critiques raised by this reviewer and is now suitable for publication in "Cells".
Reviewer 3 Report
Comments and Suggestions for Authors
Thanks for doing modification